# Global mental health and psychosocial support programming: An expert review of major implementation and funding challenges

## Perspective

culture; global mental health delivery; health inequalities

**Corresponding author:**
Paul Bolton;
Email: pbolton1@jhu.edu

*indicates shared first authorship

Paul Bolton[1]*, Saloni Dev[2]* , Ali Giusto[3], Bibhav Acharya[4],
Phiona Naserian Koyiet[5], Rabih El Chammay[6,7], Judith Bass[1] ,
Pamela Y. Collins[1] , Joseph Reginald Fils-Aimé[8] , Chenjezo Grant Gonani[9],
Erin Ferenchick[10], Laura Murray[1], Veronica Cho[11], Fátima G. Rodríguez-Cuevas[12],
Nawaraj Upadhaya[13], Giuseppe J. Raviola[14,15], Nagendra Luitel[16] ,
Esubalew Haile Wondimu[17], Inge Petersen[18,19] , Ksakrad Kelly[14], Helena Verdeli[20],
Milton L. Wainberg[10,21] , Antonis Kousoulis[22], Hamid Dabholkar[23] and Victor Ugo[24]

[1]Department of Mental Health, Bloomberg School of Public Health, Johns Hopkins University, USA; [2]Tufts University School of Medicine, USA; [3]Florida International University, USA; [4]UCSF Weill Institute for Neurosciences, USA; [5]World Vision International, Kenya; [6]Ministry of Public Health, National Mental Health Programme, Lebanon; [7]Saint Joseph University, Lebanon; [8]Zanmi Lasante, Haiti; [9]Partners in Health Sierra Leone, Sierra Leone; [10]Columbia University Vagelos College of Physicians and Surgeons, USA; [11]Global Mental Health Action Network, UK; [12]HEAL Initiative, USA; [13]HealthRight International Inc, USA; [14]Partners In Health, USA; [15]Harvard Medical School, USA; [16]Transcultural Psychosocial Organization Nepal, Nepal; [17]International Rescue Committee, USA; [18]University of KwaZulu-Natal, South Africa; [19]University College London, UK; [20]Teachers College Columbia University, USA; [21]New York State Psychiatric Institute, USA; [22]United for Global Mental Health, UK; [23]Parivartan Trust, India and [24]The MHPSS Collaborative, Denmark

## Abstract

The global mental health (GMH) field aims to equitably improve mental health and well-being everywhere. This article reviews persistent common challenges hindering sustained, high-quality delivery of mental health and psychosocial support (MHPSS). Our focus is on programming that is funded or implemented by external organizations, typically universities or international non-governmental organizations from high-income countries. It is a consensus statement of MHPSS practitioners, programmers and researchers working for these organizations and some who are locally based who observe these programs in action. We comment on progress to date, barriers and recommendations for change and the importance of promoting sustained integration of MHPSS into health and social service systems through a comprehensive, recovery-oriented system of care. We call for prioritizing often-neglected issues (e.g., stigma, severe mental health conditions and neurodevelopmental conditions), strengthening workforce training and supervision and monitoring and evaluation systems to ensure program quality. The continued dominance of the Global North in shaping GMH programming priorities remains a concern. We advocate for a greater involvement of local workers and communities in agenda-setting for programs, culturally grounded implementation and long-term capacity building. Evidence-based practices must be met with contextual relevance, and comprehensive guidelines for sustained support are needed for development settings. For persistent funding challenges, we recommend clearer funder objectives, investment in in-house mental health expertise and funder coordination with prioritization of complementary programming. These recommendations are essential to realizing equitable, comprehensive, evidence-based and contextually grounded GMH programming.



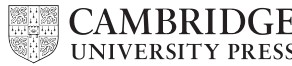

## Impact statement

Sustained, high-quality delivery of mental health and psychosocial support (MHPSS) remains a challenge in many contexts. This article is a consensus of practitioners, program managers and researchers on persisting programmatic and funding barriers and practical recommendations for change. The focus is on the funding and approaches implemented by external actors working in low and middle income countries, as these programs continue to be important contributors to mental health in many of these countries. By drawing attention to the major ongoing challenges in a single article, combined with recommendations, we intend to inform development of new and higher standards for GMH programming in development contexts, resulting in more equitable, contextually relevant and sustainable external contributions to GMH programs. Ultimately, it contributes to realizing the field's goal of a global mental health ecosystem that is accountable, inclusive and responsive to the diverse needs of the communities and populations that it seeks to serve.

## Introduction

Widespread attention to mental health as a global health issue began with the first Global Burden of Disease report in 1993 which described depression as a major cause of global disability (Berkley et al., 1993). This was followed by World Health Organization's (WHO) World Health Report in 2001 with an unprecedented focus on the large burden of mental disorders and the treatment gap driven by stigma, discrimination and lack of access to services (WHO, 2001). Global mental health (GMH) has since evolved into a broad field encompassing all actions whose purpose is to equitably improve the mental health and well-being of individuals, communities and populations in all countries. This is articulated in the definition proposed by Collins (2020), "Global mental health is an evolving field of research and practice that aims to alleviate mental suffering through the prevention, care and treatment of mental and substance use disorders and to promote and sustain the mental health of individuals and communities around the world." GMH, therefore, refers to an overarching field, including legal initiatives to repeal discriminatory laws and protect the rights of individuals with mental health conditions, policies that provide access to mental health care, advocacy and lobbying of influential stakeholders, research to expand knowledge and practice and prevention and treatment of mental health conditions. This last category is collectively referred to as Mental Health and Psychosocial Support (MHPSS; UNHCR, 2024)[1] and is the focus of this article.

This article is a review of the major implementation and funding challenges to sustained, high-quality delivery of MHPSS in global mental health practice. Our focus is on programs led by external actors, usually international non-governmental organizations (NGOs) or universities from high-income countries. We laud the progress made so far, particularly the finding that the major approaches to treatment have been effective across cultures and can therefore form a basis for programming, when appropriately adapted. We see the need for adjustments in current practices to enhance the reach and impact of these programs. Our focus is on programming that *directly*[2] promotes mental health (MH) and psychosocial well-being, plus prevention and treatment of mental health conditions. This includes information sharing (psychoeducation) and general counseling, coping skills training and environmental and behavior changes to reduce stress and clinical treatments, including medications and psychotherapies. While previous articles have critiqued specific programs (Srivatsa and Stewart, 2020), our purpose is to identify challenges we see as common across externally driven MHPSS programs in practice and make recommendations for change.

This article presents perspectives of a diverse group of MHPSS practitioners, program managers and researchers, based on our collective observations and experiences in high-, middle- and low-income countries. We acknowledge, however, that many authors – though originally from the Global South – are affiliated with US-based institutions and therefore the content should not be interpreted as representative of all GMH contexts. Given the long-term challenge of inadequate funding, it is more important than ever that the available MHPSS resources are used effectively and efficiently to best serve populations and to provide the basis for future scale-up of funding and programming. Our intent is to refer to current practices common across the outsider-led programming and funding, while avoiding criticizing any specific organizations.

Throughout the article, we use the term "local" to refer to initiatives for a population that emerge from their culturally and geographically distinct characteristics. Ideally, these initiatives would emerge from within the population, but failing that, they could refer to the country they exist in or at least from a culturally similar region, such as sub-Saharan Africa. Beyond this, we do not consider "local" to refer to different regions or the global south generally.

The article is in two sections – Programming and Funding – in which we briefly describe challenges and recommendations for change.

## Program implementation challenges

### *Challenge: while mental health is a cross-cutting issue, standalone programs are common*

Large bodies of evidence support the bidirectional negative relationships between mental health problems and physical health issues including infectious diseases, sexual and reproductive health (inclusive of perinatal health), nutrition, non-communicable diseases and early childhood development, as well as education, violence and poverty (Doherty and Gaughran, 2014). Poor mental health has been linked to worse physical health outcomes both directly and indirectly by means of reduced uptake of prevention and treatment services (WHO, 2009; Doherty et al., 2013; Doherty and Gaughran, 2014; Belkin et al., 2017; Remien et al., 2019; Collins et al., 2021; Sparling et al., 2021; Ronaldson et al., 2024; Sisodia et al., 2025). People living with mental health conditions die on average 15 years earlier than the general population (Chan et al., 2023), and yet MHPSS programs are often separate from physical health programs if they exist at all. Separation increases cost and reduces access, which is further exacerbated by the stigma surrounding mental health care. Standalone programs – programs that are operated independently of existing health and social systems –are less likely to be sustained or scaled up than those integrated into existing health and social systems. Meanwhile, programs addressing physical health, education and social problems, are less likely to achieve their objectives without addressing the contributing mental health issues.

### *Recommendation: support integration into physical health and other programs*

Integrating MHPSS into physical health and other service programs can greatly improve uptake of mental health services through easing access and ensuring privacy. Integration is cost-effective and promotes MHPSS sustainability as part of existing services (Cubillos et al., 2021). There is strong evidence that addressing mental health issues as part of these programs contributes to achieving their program goals, including programs for Reproductive, Maternal, Newborn, Child and Adolescent Health (RMNCAH) and Communicable and Non-communicable Diseases (NCDs) (Kaaya et al., 2013; Kulisewa et al., 2019; Musayon-Oblitas et al., 2019; Stein et al., 2019; Tachibana et al., 2019; Tol et al., 2020; Safren et al., 2021). For example, integrating mental health care into tuberculosis care

---

[1]The term Mental Health and Psychosocial Support (MHPSS) was first introduced through a consensus-based process involving multiple organizations including WHO that led to the IASC Guidelines for Mental Health and Psychosocial Support in Emergency Settings (IASC, 2007). However, it has evolved over time. Many actors in the global mental health space have since expanded the term to non-emergency settings, and this is how we are using it here.

[2]We exclude activities that improve mental health as a spillover or secondary effect such as poverty alleviation or employment programs because almost all programs that assist populations also contribute to mental health. We do include actions within non-mental health programs that specifically address mental health issues, for example, stress reduction counseling within programs to reduce community violence.

improves both mental health outcomes and tuberculosis treatment adherence (Sweetland et al., 2019). Such integration should also support individuals with severe mental illnesses such as bipolar disorder or schizophrenia, who are at heightened risk of neglect, abuse or exclusion from essential services, particularly when mental health services are siloed from broader physical and social services systems. Integration with social services simultaneously addresses social factors and mental health, enhancing the impact of both, for example, early childhood development and violence (Kohrt et al., 2018; McGorry et al., 2022). Poverty alleviation from cash transfer also is enhanced by addressing depression, grief, anxiety, etc. (given the bi-directional causal relationship between poverty and mental health), otherwise running the risk of worsening inequity by leaving behind those with these issues (Ridley et al., 2020). The impact of cognitive behavioral therapy – a common treatment for depression – can be enhanced by economic support among poorer populations (Blattman et al. 2017). While the recommendation to integrate mental health with physical health and social services is not new, particularly in humanitarian settings (IASC, 2007), it needs to be implemented more systematically in practice. We advocate for such a comprehensive approach in settings facing longstanding emergencies as well as developmental contexts.

Integrating mental and physical health care should be done through a collaborative care model which helps identify physical conditions with psychological or behavioral symptoms and vice versa, reducing the risk of diagnostic overshadowing (Firth et al., 2019; Ee et al., 2020). Integration means a comprehensive care system of inter-connected and complementary interventions to address the full spectrum of need, encompassing both psychosocial support and mental health treatment. The former includes prevention efforts such as changes to the environment to reduce stress (by engaging employers,[3] social services, local governments and private groups), as well as integrating mental health education and stress management as part of adult education and school curricula. The latter – treatments for mental health conditions – includes community-based care linked to consultation and referral to professional mental health providers and temporary hospitalization where necessary. Addressing mental health problems through integration at the primary care level decreases need for referral to expensive and often inaccessible professional mental health services including psychiatric hospitals. Principles derived from implementation science can guide enhanced integration. For example, in Nepal, Rimal et al. (2021), used the capability, opportunity, motivation and behavior (COM-B) framework to improve primary care providers' treatment skills, motivation, access to counselors and psychiatrists, and a reliable medication supply. The program has been scaled up to serve over 10,000 beneficiaries in Nepal and is a model that the WHO is using to field test mental health integration in Ghana and Dominican Republic. Some authors on this article and colleagues have outlined the comprehensive, collaborative and community-based care or "C4" framework for the integration of stand-alone mental health programs with health and social community-based services. It delineates the types of workers involved, treating comorbidities as a norm and explicitly addressing referral and collaborative pathways (Bolton et al., 2023).

Community-level treatment providers should be trained in multi-problem, modular methods as well as transdiagnostic approaches to address comorbidities (Lovero et al., 2021).

Addressing common mental health conditions such as depression, anxiety and trauma together produces greater impacts on each of these conditions than treating them separately. Such an approach is also more scalable, broadly relevant to community needs (Wainberg et al., 2017; Kane et al., 2018) and aligns better with current "real-world" clinical practice (i.e., how clinicians treat clients). Mozambique, which faces a critical shortage of health resources, has adopted some of these innovations. For example, the country has formalized task-sharing within its national health platform to meet multiple population mental health needs (Dos Santos et al., 2016), which has led to the sustainable scale-up of comprehensive mental health and substance use care that has been integrated within their community primary care clinics system with existing human resources (Wainberg et al., 2021).

Digital technology has become a promising tool in integrated health care, particularly mental health care addressing depression and anxiety in low- and middle-income countries (LMICs). It offers a cost-effective solution that enhances non-specialist providers' capacity, supports task-sharing of evidence-based practices (EBPs) and makes mental health care more accessible. Mobile-delivered brief EBPs improve treatment rigor and fidelity, providing comparable outcomes to traditional methods while reducing costs (Naslund et al., 2017). Digital psychosocial interventions and training for non-specialist workers in LMICs show moderate effect sizes – comparable to those in high-income countries – for conditions like depression and substance misuse (Fu et al., 2020). While effectiveness against treatment as usual varied due to differing standards of care, digital interventions showed strong effects compared to no treatment, highlighting their potential in settings with limited mental health services. Some examples of effective digital interventions are as follows: Inuka, a chat-based application based on the Friendship Bench problem-solving therapy intervention (Dambi et al., 2022), and Step-by-Step, a digital tool based on behavioral activation to treat depression (Cuijpers et al., 2022). Additionally, digital platforms enable record-keeping and remote monitoring of patient progress. Given these benefits, integration of digital mental health interventions, particularly in LMICs, is recommended to strengthen mental health care provision and increase access to care. The Red Cross Red Crescent (RCRC) Movement MHPSS Hub (2024a); b; c), for example, has published resources to guide the development of human centered digital MHPSS solutions that can be utilized in relevant settings. Large-scale implementation trials are needed to assess the long-term effectiveness and sustainability of these interventions in diverse contexts (Wainberg et al., 2021).

Each community needs to be part of such a system of MHPSS integrated into their existing and commonly accessed health and social services (WHO, 2025a). Each point of integration is also an assessment and triage point through which clients can access whatever combination of services they need. Each model recognizes the need to meet people where they are in terms of needs and resources and provide a mechanism to "step up" care for those with greater needs and/or comorbidities. In Liberia, for example, a Universal Health Coverage pilot is implementing cross-sectoral engagement and sustainable change for integration of mental health with NCD care through collaborations with persons affected, formal and informal healthcare sectors and multidisciplinary technical advisory boards (Godwin-Akpan et al., 2025).

Successful integration necessitates addressing workforce challenges. Most local MHPSS programs, including programs integrated into other programming, rely on task sharing – the sharing of treatment and psychosocial programming skills with non-professional

---

[3]Workplace mental health was the theme of 2024 World Mental Health Day (WHO, 2024b). See also, Guidelines on Mental Health at Work (WHO, 2022).

workers. This emulates the primary health care model of training non-professional health workers in specific skills most in need. At the local level, existing workers can incorporate psychosocial skills to their roles so long as they have the training, supervision and dedicated time and space to do so, while full treatment with counseling therapies requires dedicated workers (see, section on program quality below). The right combination of existing workers who take on psychosocial support activities alongside their other responsibilities plus staff specifically hired to provide mental health treatment will depend on local programs and capacity (Bolton et al., 2023). Integration also requires linking the health information systems so that clients are tracked across both MHPSS and other health care. Making this work and truly integrating mental health as a cross-cutting issue across services requires support from not just the ministry of health but also from other ministries related to women's issues, child protection and finance.

### Challenge: programs often address only a few issues rather than the range of mental health and psychosocial problems of those most in need

While there has been extensive research and development of interventions to address common mental health disorders such as depression, anxiety, posttraumatic stress, as well as generalized distress, other important mental health issues have received insufficient program attention:

○ Stigma and discrimination against persons with mental health conditions
○ Treatment of severe mental health conditions, such as bipolar disorder and schizophrenia.
○ Safety against self-harm or suicide
○ Youth and child mental health conditions and neurodevelopmental conditions
○ Substance misuse and abuse
○ Intimate partner violence (IPV) and gender-based violence (GBV) prevention by addressing the mental health of both victims and perpetrators.
○ Variations in mental health issues by gender and sexuality

Programs do not often address stigma and discrimination, yet these are universal and affect all levels of society. They are major barriers to individual, collective and societal action toward addressing mental health, affecting prioritization, resources, quality of programs, service delivery and service seeking. They are based on widespread misconceptions and fear: that mental health conditions is a personal failing or an inherited weakness, that persons with mental health conditions cannot hold responsible positions or healthy relationships, that treatment is expensive, long term and relatively ineffective. Multiple reviews of demand-side research link lack of help seeking to stigma and poor understanding and awareness of effective services (van den Broek et al., 2023; Dumke et al., 2024). Stigma and discrimination produce marginalized, hidden populations of individuals with mental health conditions who are reluctant to seek care and unable or unwilling to advocate for their own needs and interests.

While treatment of severe mental health conditions is often the sole focus of existing routine mental health services in many countries, there have not been efforts to make improvements or link them to community-level services. MHPSS programs led by external, academic or international NGOs instead tend to provide, per our definition, psychosocial support only in that they focus on reaching large numbers of relatively accessible people with interventions focused on prevention of single conditions (e.g., perinatal depression). Issues perceived as more difficult or expensive – treatment of mental health conditions, safety, programs for substance abuse and programs intended to reach youth and marginalized groups – are less often programmed. Where programs do offer treatment, it frequently targets a small number of conditions (such as depression or trauma) despite comorbidity being more common (Martel et al., 2017; McGrath et al., 2020; Giusto et al., 2022) particularly in conflict-affected areas (Charlson et al., 2019). For example, Mental Health Gap Action Program (mhGAP; WHO, 2016) uses algorithms suited to diagnose and treat mental health conditions in isolation. Its algorithmic and medication-based approach is difficult to apply to the treatment of comorbidity. Therefore, non-specialists with only a brief mhGAP training are not equipped to manage comorbidities, unless they also reliably receive ongoing supervision from specialists such as psychiatrists or are trained in transdiagnostic psychotherapies.

Programs do not consider the influences of gender and sexuality on mental health. Mental health conditions tend to be more frequent among women, partly due to domestic and gender-based violence and lower status in many societies (Kiely et al., 2019). Men have higher suicide risk and higher rates of substance use but are less likely to seek care due to social expectations that they be resilient, due to the pressure to earn income as the main supporters of their families, and potentially other factors that are under-explored. Currently, the most frequent approach to addressing gender and sexuality in mental health is to limit programming to a single gender, usually women. Similarly, schizophrenia, bipolar disorder, substance-abuse, post-traumatic stress disorder, etc., also place a significant disability burden on a subset of the population, yet community- and primary-care-based mental health programs tend to overlook these illnesses.

While social determinants are an impact factor in mental health, they are not identified as targets for MHPSS programs. The very limited funds for MHPSS programming require a focus solely on the provision of MHPSS services rather than on determinants like poverty and education. With poverty being a subject of a much larger existing programming, recent evidence suggests that poverty-reduction interventions combined with psychological interventions frequently drive significant mental health gains, as compared to psychological components alone (Tanski et al., 2025). Integrating MHPSS with those that address social determinants can also expand the reach of mental health services toward universal access (WHO, 2025a).

### Recommendation: programs need to address these issues as part of comprehensive care – both in terms of addressing all major mental health problems and barriers and reaching all those in need

Stigma and discrimination need to be addressed in MHPSS programming at all levels: patients, their families, communities, health-care workers, counselors, leadership and policy makers. This includes mental health literacy programs for the general population to correct misperceptions and provide accurate information on mental health care services (WHO, 2025a). Enabling people to be more open about their situation, especially those experiencing severe mental illnesses, is also essential as there is ample evidence that social contact with people with lived experience is effective at reducing stigma (Thornicroft et al., 2022). This requires addressing systemic stigma and discrimination, which is best done empowering individuals with lived experience to engage in creating safe

social environments, co-developing and implementing needs assessments, programs and research and in the design of service provision (WHO, 2025a). Increasing their visibility, leadership and engagement in implementation, in turn, requires addressing structural discrimination by governments, employers and other institutions, to repeal discriminatory laws and policies. These should be replaced with protections for the rights of individuals with mental health conditions, drawing on existing human rights frameworks (UNGA, 2006) and instruments that promote equitable treatment through focus on recovery and rehabilitation into the wider society.[4] These are necessary to engage political stakeholders and garner community support while encouraging and assisting people with mental health conditions to use services.

Programs should include gender and sexuality considerations in staff training (e.g., sensitization on gender equity, trauma-informed approaches and GBV prevention), monitoring and evaluation (e.g., disaggregating indicators by gender) and intervention design (e.g., gender-specific strategies for expanding access).

### Challenge: mental health programming priorities and practices in LMIC continue to be dominated by the global north, with insufficient local involvement and adaptation

The global mental health field has been described as one of north to south power imbalances and colonialist practices (Rivera-Segarra et al., 2022). Despite progress toward equity, imbalances remain (Abimbola and Pai, 2020). Western, well-resourced institutions (e.g., NGOs outside of the local or national context) have dominated global mental health with less input from the Global South, including local policy makers, providers, patients and other community voices. This has resulted in widespread adoption of Western concepts and methods of screening, diagnosis, and treatment (Moleiro, 2018). While cross-cultural research has supported the widespread applicability of these approaches, it also emphasizes the need for local adaptation, yet they are often not well adapted and only partially reflect local mental health issues. For example, in societies where collectivist values, spiritual beliefs and traditional healing methods can also play a central role in mental well-being (Bass et al., 2007).

Most mental health assessment by external actors relies on Western instruments, which, in turn, are based on symptom diagnostic criteria drawn from the Diagnostic and Statistical Manual of Mental Disorders (DSM) and International Classification of Diseases (ICD). One argument for using these instruments without adaptation is the need for standardization of assessment across populations. But using the same instrument across populations can result in a lack of comparability. For example, if some aspects of depression manifest differently in different populations using the same instrument may result in not measuring depression correctly in any of them. This can go deeper: western assessments prioritize symptoms as measures of need and impact, with less attention to broader challenges. This includes the impact of mental health problems on key activities of daily living, particularly those that are necessary to support self and family, which are often higher priority locally. Externally developed instruments are often poorly adapted and not well validated locally. For example, the Patient Health Questionnaire (PHQ-9) is widely used but often without local adaptation and validation in the populations surveyed.

The same considerations relate to treatment, which also cannot be just the importation of Western approaches. Most successful treatment approaches have undergone significant adaptation of their materials to be acceptable, implementable and sustainable. Such local adaptation is not the norm for most programming.

### Recommendation 1: emphasize locally driven design and implementation with North–South cooperation and capacity-building

This should be done under the rubric of mutual capacity building, defined as an equal bidirectional exchange of ideas between LMICs, Small Island Developing States (SIDS) and high-income countries (HICs) to promote shared learning toward increasing local system appropriateness and capacities (Wainberg et al., 2007; Binagwaho et al., 2013; Jack et al., 2020; Giusto et al., 2024; Jack et al., 2024). This requires fostering truly mutual and equitable partnerships in which all partners reflect on each aspect of programming from the definition of concepts and goals to implementation and choice and measurement of indicators and interventions (Wainberg et al., 2007; Bemme et al., 2024). It includes adopting evidence-based interventions and guidelines from the Global South in the Global North in ways that move beyond "reverse innovation" (Ngaruiya et al., 2021) and require the same process of local adaptation as North to South. For example, a program developed in the Global South to improve depression would not just be adopted by a community in the United States, but undergo appropriate adaptation and testing to ensure cultural relevance (Giusto et al., 2024) as well as effectiveness. Mutual capacity building encourages communication and partnership across sites to build capacity to improve mental health in both settings. This shift increases input from local policy makers, providers, patients/individuals with lived experiences and community voices.[5]

Mutual capacity building is a formal process that requires more than initial ad hoc interviews with local stakeholders to inform program design. Community-based participatory research (CBPR) methods provide structured and in-depth approaches to collaborative decision-making, centering the voice of communities in program design and implementation (Wainberg et al., 2007; Wallerstein, 2021; Payan et al., 2022). CBPR is part of the broader Community-Engaged Research (CEnR) approach which emphasizes *ongoing* engagement of the community as experts, shared decision-making and emphasizes shared values, goals, communication processes and power. It incorporates specific strategies for community engagement such as creation of meaningful advisory boards with community representation, as well as principles that center reflection, clear communication, honesty and mutual respect (Chou and Frazier, 2020; Goodman et al., 2020). Implementing organizations should routinely embed this type of structured approach to collaboration for program implementation. Funders, who play a critical role in sustaining mental health programming, could use this approach to determine priorities and funding cycles.

A participatory approach of local adaptation is necessary even with existing globally recognized approaches. For example, the Nepali version of the mhGAP Intervention Guide includes additional modules for anxiety disorders and conversion disorders to meet local needs (Luitel et al., 2020). Interpersonal Psychotherapy has proved

---

[4]WHO's QualityRights Initiative provides training to improve the quality of care in MHPSS services and to promote the rights of people with psychosocial, intellectual and cognitive disabilities (Funk and Bold, 2020).

[5]A baseline knowledge across teams around historical colonialism (i.e., the one-sided extractions of resources from the Global South) can be beneficial in framing discussions and avoiding past unhelpful practices (Abimbola and Pai 2020; Büyüm et al., 2020; Kim, 2021; Sweetland et al., 2016).

effective in various cultures, but only after local adaptation by providers (Verdeli et al., 2003). We acknowledge that such adaptations could be challenging amidst a lack of time, resources and clear guidance. However, participatory adaptation is necessary and feasible where there is already a deep understanding of traditional practices. In our rapidly evolving world, traditional practices and beliefs have often been eroded, particularly among the young. So, in each new setting, there is a need to conduct pre-assessments before program design, understanding the perspectives of all groups to be affected.

### Recommendation 2: recovery should be the primary objective of treatment services and, therefore, a major part of impact assessment. Instruments that assess recovery and symptoms need local adaptation and testing prior to use

Recovery should be the main outcome of mental health care services (WHO, 2025a). Recovery is "a process of change through which individuals improve their health and wellness, live a self-directed life and strive to reach their full potential" (Substance Abuse and Mental Health Services Administration (SAMHSA), 2014). This places a focus on activities required for a satisfying and meaningful life. While standard functioning assessment instruments exist and can be a resource, like the WHO Disability Assessment Schedule (WHO-DAS 2.0), they require local culturally appropriate adaptation to be relevant and sensitive to important local activities. Western functioning instruments still rely on individual productivity impairment whereas a broader global view of functioning includes social wellness and collective capabilities (Bolton et al., 2004; Hartman et al., 2023). Knowing what activities and symptoms are important locally requires knowledge of the local community which requires meaningful collaboration with community members using approaches such as CBPR, as previously described. Symptoms should continue to be assessed, with the inclusion of those known to be relevant and distressing locally.

Screening instruments and other measurements need to be locally adapted. Rather than thwarting cross-cultural comparison, this enhances it. For example, only if an instrument truly assesses the local manifestation of depression can the results be compared with assessments of depression in other populations. This requires a prior understanding of mental health and how it manifests in each population, which in turn requires prior qualitative investigation, which is still infrequently done.

Screening instruments and other measurement tools also require local validity and reliability testing to determine how well each item, and each instrument as a whole, is performing. Poor performance, particularly by individual items, is common even with instruments that have undergone local adaptation. Instrument adaptation and testing need to be a standard part of program development unless proven instruments already exist for the same population. An example is the Cognitive Processing Therapy (CPT) Functionality Tool to assess and monitor the impact of CPT on individuals recovering from trauma (Elhra, 2014).

### Challenge: program quality is not well supported

Program quality is often poor due to inadequate training and supervision, a lack of monitoring and impact evaluation and inadequate integration of workers and services into existing health and social systems. Interventions too often have little or no scientific evidence of effectiveness relevant to the populations being served, yet are often transplanted from one global region (often Western) to another with little adaptation or local impact assessment.

While guidelines and manuals for some evidence-based low-intensity psychological interventions are accessible online, operational challenges such as inadequate training, lack of cultural adaptation and limited supervision affect the quality of their frequent "off the shelf" use. The alternative approach – direct training – tends to have more thorough processes for training, supervision and fidelity, but otherwise is affected by the same issues. Direct training also faces challenges with a lack of trainers and their time and can involve high costs when trainers are based in HICs.

Insufficient resources, particularly workers and worker time, are a significant challenge. MHPSS services in LMICs often exist solely as "add-ons" to the duties of existing and already busy staff without additional paid time or resources. Lower intensity psychosocial or brief psychological interventions (e.g., brief behavioral activation delivered over 2–4 sessions or single session interventions) are important first-line services that can be delivered as add-ons by community health workers (CHWs) or other community-based workers, but only if these workers have clear and substantial time carve-outs to provide these services. Full psychotherapeutic services of 8–12 sessions (e.g., cognitive behavioral therapy (CBT), interpersonal psychotherapy (IPT)) are more time-consuming and require different skill sets from those of most existing program staff, including CHWs and physical health providers. They cannot be added to existing job descriptions, instead requiring dedicated psychotherapeutic workers working full-time.

Monitoring and evaluation are critical to maintaining the quality of key program elements: assessment and triage, supervision, fidelity, effectiveness and provider and client satisfaction. While task-sharing has been shown in trials to be an effective method of providing MHPSS services (Lange, 2021), quality of services is not sustainable without systems of continuous mentorship, supervision and support. Supervision is of prime importance, since assessing and maintaining quality is a vital function of good supervision, particularly for services provided through task-sharing. Supervision also provides the necessary ongoing training.

### Recommendation: expand focus on training, supervision and monitoring and evaluation, providing the necessary time and resources

A primary consideration for program quality is ensuring that workers receive training, provide services, participate in supervision and support monitoring and evaluation through dedicated funding, time and resources, as described above.

Most task-sharing training is an add-on and ad hoc for existing workers only. Including mental health training in pre-service training (e.g., part of curricula at academic/training institutions as recommended by WHO's guide on pre-service education (WHO, 2025b)) or part of existing in-service training to expand existing skills or add new ones is less expensive than postgraduate or post hoc training and can improve professional standards and quality of care. Professional schools are more capable of assessing, improving and confirming the competencies of counselors compared to ad hoc sessions at the workplace. Integration into professional training helps inculcate mental health as an important and routine aspect of the role of existing workers.

In-service training can include short courses, but the most essential components to building competency are supervised practice followed by ongoing supervision to refine skills, support workers' own mental health and sustain motivation (Wainberg et al., 2021). For mental health specialists in LMICs, they need to master the skills needed to work within task-shared interventions: a)

working in teams, b) collaborative clinical decision-making, c) supporting the quality of team members' work, d) overseeing a large panel of patients to ensure people are not falling through the cracks; e) providing supervision for non-specialists to deliver mental health care rather than directly delivering care (Acharya et al., 2017; Jackson et al., 2022). The case conference is an effective supervision method where trained primary healthcare workers gather, led by a psychiatrist or other professional involved in training these workers. Workers present patients and receive feedback from other health workers and the trainer. This approach has been reported as helpful for both healthcare workers and patients (WHO, 2018a).

Workforce development and maintenance require respectful integration of mental health workers as part of the health and social services systems. Each worker needs a title and pay appropriate to their training and work. This should be a program prerequisite, particularly for task-sharing approaches. Otherwise, when organizations hire, train and support lay workers, but without official integration into existing systems, their roles disappear when funding dries up.

Each intervention should include fidelity measures that can be used to train, assess and retrain workers to ensure that their services are high quality. Existing evidence-based treatments (e.g., CBT, IPT, narrative exposure therapy (NET)) tend to have fidelity monitoring incorporated in their supervision and certification processes. For interventions that do not, the Ensuring Quality in Psychological Support (EQUIP) model is a WHO and UNICEF project that aims to enhance the skills of support providers while ensuring training and service delivery are consistent and high-quality. EQUIP consists of a platform with free competency assessment tools and e-learning courses to support governments, training institutions and non-governmental organizations, both in humanitarian and development settings, to train and supervise the workforce to deliver effective psychological support to adults and children (Kohrt et al., 2020; Kohrt et al., 2025).

### Challenge: interventions are selected without reference to the best available evidence

MHPSS programs frequently include interventions that are not adequately evidence-based[6] for the populations and problems in question, or even similar populations. While evidence is lacking for most interventions in most populations, it is critical that we make the best use of the best available evidence. WHO's mhGAP (WHO, 2016) recommends evidence-based therapies treatments such as the Thinking Healthy Program (WHO, 2015), Interpersonal Psychotherapy (WHO and Columbia University, 2016) and psychosocial interventions developed by WHO such as Self-Help Plus (WHO, 2021b) and Problem Management Plus (WHO, 2018b). Yet, program decisions are often made at a high and external level by professional administrators and/or by external mental health experts, often from a HIC and with the lack of the local input described previously. External technical experts often base their selection of interventions on that which they are familiar without

reference to existing studies and prior programming among relevant populations.

### Recommendation: promote the adoption of evidence-based interventions by reference to continuously updated databases of interventions

Decisions on the content of programs should be based on consultation of up-to-date global databases of evidence-based interventions that include what counterfactual studies have been done, among which populations, and for which problems. Criteria for relevance of evidence, from highest to lowest, could be: (1) Studies conducted among the population in question for the same issues to be addressed; (2) Studies conducted among similar populations for the same issues; (3) Studies not conducted among the same or similar populations but among multiple diverse populations; (4) Studies conducted among a single population different from the population in question (Groves et al., 2021). Existing updatable databases that have much of this information include those created by the Mental Health Innovation Network and Metapsy.

While intervention choice should be based on scientific impact, there are still places in the world for which no relevant scientific evidence exists for any specific intervention. In these situations, collaboration with local providers, patients and community voices (via a CBPR approach) to gain insights into what is accessible, appropriate and acceptable is an appropriate basis for selection. The mhGAP Community Toolkit could also serve as a resource for program managers to identify local mental health needs and tailor services to address these needs. However, implementation should include impact assessment, preferably a counterfactual study, to confirm effectiveness.

### Challenge: lack of guidelines for development contexts

While many countries have developed their own mental health systems and guidelines, many have not. For those without local guidelines, major international organizations, including the United Nations International Children's Emergency Fund (UNICEF), WHO and the UNHCR, have taken a lead in establishing guidelines for MHPSS programs. Primary among these are the development of the Inter Agency Standing Committee (IASC) Guidelines for Mental Health and Psychosocial Support in Emergency Settings (IASC, 2007), World Health Assembly's (WHA) Resolution WHA77.3 on "strengthening mental health and psychosocial support before, during and after armed conflicts, natural and human-caused disasters and health and other emergencies" (WHO, 2024a). Development programs are often based on these guidelines, which, while providing excellent guidance for humanitarian contexts, were not designed to support long-term system development. While they do emphasize coordination with development platforms, as well as "building up" the MHPSS system and referral mechanisms, their humanitarian focus means they lack detailed guidelines on establishing sustainable, locally integrated mental health systems.

WHO has taken a leading role in mental health service implementation in development with the mhGAP program (WHO, 2008). Treatment under mhGAP is focused on diagnosis of single mental health conditions by primary care providers and treatment by medications, which WHO acknowledges are overall less effective than counseling therapies for common mental health conditions, particularly trauma. In 2024, WHO added a manual for integrating counseling/psychological interventions to fill this gap (WHO, 2024c). However, the manual refers to the integration of one or a few

---

[6]Here "evidence-based" refers to: 1) counterfactual studies conducted among the same or similar population for the same or similar problems as the proposed program. Given the many varying factors that affect mental health only a counterfactual study can determine the impact of the program compared to not providing it. 2) These or related studies need to also explore local implementation, to ensure that programs are locally acceptable, appropriate, accessible, feasible, scalable and cost-effective.

interventions for a limited number of specific conditions, rather than a system of care meeting the various and usually comorbid mental health problems of individuals and populations. The manual is also brief and necessarily nonspecific, functioning more as a "what to do" rather than a "how to" for integrating a system of mental health care that meets all the major needs of people at all ages. WHO has also produced resources and toolkits for implementing mental health services in primary care (WHO, 2023a) which provides valuable tools and serves as a reference, but a focused guidance document for program development has not been produced.

### Recommendation: develop comprehensive guidelines for mental health for the development sector

We need specific implementation guidelines for integrating a comprehensive mental health system into development programming. This system needs to incorporate community-based prevention, promotion and treatment linked to professional care as needed. The guidelines need to provide clear, actionable direction rather than just a broad list of issues to consider, drawing on existing resources from WHO but going further to provide concrete guidance for implementation. Guidelines for development and humanitarian response should reflect the transition from relief to development, rather than operating without consideration of the other. This can ensure that emergencies and disasters make use of opportunities provided by short-term expansion of attention and resources for permanent improvements in mental health services and policy. For example, in Lebanon, the Ministry of Public Health capitalized on enhanced global interest and funds following the Syrian influx to reform mental health policy and systems and build long-term services for all persons (host population and the displaced) who reside in Lebanon.[7] Post-earthquake recovery and rebuilding in Haiti provided a similar opportunity. Both of these cases have been described, alongside a proposed list of competencies for humanitarian mental health responses (Raviola et al., 2023), but the resulting comprehensive guidelines have not yet been produced.

### Funding

### Challenge: funding levels are far below what is needed or warranted by the impact of mental health on global crises and development

As of 2024, the global median of public spending on mental health was just 2.1% of the government health expenditure (WHO, 2025c). Considerable progress has been made in the last few decades, with 63% of WHO member countries (89% of respondents to the 2024 World Mental Health Atlas survey) reporting having a mental health policy or plan. However, of the latter, only half have estimated the financial resources that they may need, 75% have included an estimate of human resources required and only 16–18% have fully allocated the needed financial and human resources (WHO, 2025c). External partners try to fill these financing gaps through development assistance for mental health (DAMH). Development organizations provided over $200 million in DAMH in 2021, with private/philanthropic donors providing over half of the DAMH and bilateral donors around 36–41%. This was a

decrease of $100 m from 2018 (United for Global Mental Health, 2023). At this time, 2025, the available funds are much less due to the dismantling of USAID and large reductions in foreign assistance by other countries, including major donors – The United Kingdom, France, Canada and The Netherlands.

Even before the 2025 cuts, low levels of funding for mental health programming had become common, resulting in the programming gaps described previously: lack of mental health treatments and attention to some priority topics, poor training and supervision, lack of accurate monitoring and evaluation and lack of local engagement and planning. Other funding issues are inadequate coverage of mental health by insurance or government schemes, with poorly functioning referral pathways. Unfortunately, both government and external donors have come to view this as a norm – that MHPSS programming requires significantly fewer resources than physical health. There has been a "race to the bottom" where poor funding has led to programs existing on little funding, creating the expectation among funders and governments that these limited programs are appropriate, leading in turn to low funding expectations for future projects.

### Recommendation: establish guidelines on the content and approach of appropriate GMH programs

A major driver of this problem is the previously described lack of guidance on the essential components of an effective MHPSS program and how to implement them, leading funders to accept incomplete programs. Lacking programming guidelines, funder objectives and expectations are unclear, resulting in inadequate ad hoc investments in mixed quality piecemeal programming. A common issue is that funders often prioritize impacting the greatest number of people by focusing on cheaper and more easily scalable psychosocial interventions rather than meeting the needs of the most affected by funding more expensive treatment options. Guidelines are necessary that describe the necessary comprehensive PSS and MH treatment services, with realistic resources and timelines based on current evidence and local collaboration.

### Challenge: funding is given to organizations without in-house mental health expertise or local experience

Funding for MHPSS is often a subsection of much larger awards focused on physical health or, less commonly, other major issues such as education. Awardees are often chosen based on experience with these broader topics, but without mental health experience or expertise. As a result, these organizations often subcontract mental health programming to organizations in HICs with little expertise in global mental health, or in the region in which the program is being implemented. In the name of cost-saving and task-sharing, MHPSS tasks may even be delegated to personnel without appropriate training and supervision because of a lack of appreciation that such tasks indeed require technical expertise and training. Additional barriers to local grantees include English dominating grant and proposal writing, with grant-specific resources less accessible to underserved regions (Boum II, 2024), inequitable partnerships between international and local organizations (Ginsbach and Erondu, 2021), short-term project cycles, rigid funding restrictions posing programmatic limitations (Ingram, 2022), as well as the perennial problem of a misalignment of donor and national funding cycles. Collectively, these challenges undermine building effective, appropriate and sustained services.

---

[7]See the timeline for mental health reforms in Lebanon amid crises: https://drive.google.com/file/d/1kK2vLNYNCMXitDbGaPY7NyUYGRy9VW6E/view

### Recommendation: both funder and grantee staff need to have in-house mental health programming experience, with grant opportunities more accessible to local organizations

Where local organizations are the grantee, but local mental health expertise is unavailable, international or regional experts may be sought with an explicit goal of building sustainable local capacity, including mentoring local mental health leaders. Funders need to be aware of, and make accommodations for, language, local funding cycles and other local structural considerations, to facilitate applications by local organizations.

### Challenge: lack of attention and resources by governments

Lack of a strategic mental health plan with clear service frameworks and dedicated funding hinders advocacy for comprehensive services. This results in fragmented, ad hoc mental health interventions as implementers struggle to fill the gaps with limited resources. Low prioritization of mental health is also reflected in internal structures: within relevant government ministries, the mental health infrastructure is typically small, often consisting of overstretched junior officials tasked with other activities considered higher priority. Particularly concerning is a lack of ongoing health workforce development and support. While task-sharing approaches have been widely adopted based on the existing evidence (Raviola et al., 2019; Le et al., 2022) these efforts will not be sustainable or of quality without broader workforce initiatives such as strengthened training and supervision mechanisms and more attention to career development pathways. Frontline providers on whose shoulders the field proposes to transform mental health globally are typically not well paid, developed professionally or given a permanent role within existing service systems. In many countries, there are no career development pathways in mental health care delivery and certification is an issue for non-specialists such as community health workers. While external funders can assist, in the longer term, local governments need to take responsibility to build the supporting infrastructure.

### Recommendation: governments and external funders work together to create internal structures to implement and budget permanent mental health services

Mental health cannot receive appropriate attention and resources when responsible government entities lack a plan, structure, support and resources. External actors can assist by reaching out to leaders in relevant government ministries to support planning and the creation of internal structures with the authority and necessary administrative and financial support to build and maintain solid service systems and with a mandate to govern the system and coordinate stakeholders (WHO, 2023b).

### Challenge: funding of programs is poorly coordinated

Some populations receive overlapping services using competing approaches, while other populations and their needs go unaddressed. For example, in Ukraine, the WHO counted more than 300 NGOs providing MHPSS services, whereas reports for Sudan, with a larger population, find less than 10 (Center for Strategic and International Studies, 2023). Similar inequalities occur within programming: as described above, multiple programs provide PSS, with few providing mental health treatments, or multiple programs may focus exclusively on a single subpopulation, such as women or children only.

### Recommendation: funders need to improve coordination and prioritize complementary programming and equitable coverage of populations in need

Coordination is required at both the headquarters level in terms of selection of regions and countries, and at the country/local level regarding types of programming and subpopulations. Funders need to communicate with each other to avoid duplication of efforts and gaps. They also need to avoid making promises that are not fulfilled, as this can result in other funders moving their support elsewhere, creating gaps that would otherwise have been filled.

### Summary and conclusions

This article is a review of MHPSS programming globally led by actors outside of the local or national contexts, based on the experience and viewpoint of the authors. We note substantial progress. Mental health considerations are now commonly included in both humanitarian and development programs and recognized as a human right (United Nations General Assembly (UNGA), 1948; 1976; 2006). The global health, social and economic significance of mental health is increasingly recognized, and more than half of United Nations (UN) Member States have now developed and adopted mental health policies and strategies. This awareness was recently heightened by the COVID-19 pandemic, where the disease, control measures, and inequities gave rise to persisting mental health challenges.

While noting this progress we have pointed out some areas for improvement as summarized in Table 1. The timing is appropriate for although governments, practitioners and external funders have become more aware of the importance of mental health, public foreign assistance funding available for mental health and even health and development in general, has recently been greatly curtailed, making it more important than ever that we make the best use of available funds.

Underlying the comments and recommendations in this article are two major issues. The first is that, more than ever, funding falls far short of reflecting the need and the importance of mental health. Efforts to do much with little are skewing programs toward relatively cheap prevention, often at the expense of comprehensive care and treatment. While prevention is important, there is also the need for effective treatment of individuals with moderate to severe mental health conditions, who are those with the greatest need.

The second major issue is a lack of widely agreed and adhered to guidelines for programming and training, particularly in the development context. Over the last few decades, diverse fields, including humanitarian assistance and child survival, have transformed their programming through agreed standards and guidelines (e.g., SPHERE Association, 2018). MHPSS needs to do the same to achieve strong, sustained systems of accessible, evidence-based care that address the spectrum of mental health needs of communities, including marginalized groups. WHO and other leading institutions have generated documents and guidelines that imply principles and standards for global mental health services, but are not explicitly stated and remain as soft recommendations without implementation guidance. Some groups have begun to generate separate statements of principles and to promote their adoption, such as the US Global Mental Health Alliance (2024) and the US Agency for International Development (USAID), which, prior to its demolition by the current US administration, produced a position paper on mental health (USAID, 2024). These, along with the WHO materials, address the "what" of programming, but no one

**Table 1.** MHPSS programmatic challenges and recommendations

| Challenge | Recommendation |
|---|---|
| *Programming* | |
| 1) While mental health is a cross-cutting issue, standalone programs are still common. | Support integration of mental health programming into existing physical health and social support programs. |
| 2) Programs often consist of a few interventions, not focused on those most in need. | An integrated comprehensive system of mental health programming that addresses the range of mental health needs across populations or subpopulations and barriers to care. Address gender differences. Address stigma, discrimination and other demand-side barriers. |
| 3) Globally, mental health priorities and practices continue to be dominated by the Global North with insufficient local involvement. | Emphasize locally driven program design and implementation with north–south partnership and sustainable capacity-building. |
| 4) Program quality is not well supported. | Expand focus on training, supervision and monitoring and evaluation, ensuring the necessary time and resources to build strong, sustainable systems. |
| 5) Interventions are selected without reference to the best available evidence. | Promote the adoption of evidence-based interventions by referencing continuously updated databases of interventions. |
| 6) Overfocus of assessment on externally generated lists of symptoms. | Recovery should be the primary objective of treatment services and, therefore, a major part of impact assessment. Instruments that assess recovery and symptoms need to be locally adapted and tested prior to use. |
| 7) Lack of implementation guidelines for development contexts. | Develop comprehensive guidelines for mental health in development contexts (i.e., low resource environments). |
| *Funding* | |
| 1) Funding levels are far below what is needed or warranted by the impact of mental health on global crises and development. | Establish guidelines on the content and approach of GMH programs that define appropriate programming and the required resources. Fund these programs as indicated by the guidelines. |
| 2) Funding given to organizations without in-house mental health expertise or local experience. | Both funder and grantee staff need to have in-house mental health programming experience, with grant opportunities more accessible to local organizations with local mental health experience by addressing structural barriers to applications and from LMIC organizations. |
| 3) Lack of attention and resources by governments. | Work with governments to create internal structures to implement and budget permanent mental health services. |
| 4) Funding of programs is poorly coordinated. | Funders need to improve coordination and prioritize complementary programming and equitable coverage of populations in need. |

has yet produced guidance on the "how", resulting in the absence of or inconsistent implementation.

Specific principles and guidance will encourage widespread consensus and adoption of effective programming, supported by adequate funding. This will move the field not only toward more effective use of existing funds but also provide a programmatic basis to advocate for expanded resources.

**Open peer review.** To view the open peer review materials for this article, please visit http://doi.org/10.1017/gmh.2025.10093.

**Data availability statement.** Data availability is not applicable to this article as no new data were created or analyzed in this study.

**Acknowledgements.** We sincerely thank Dr. Manasi Kumar for her valuable feedback on this manuscript. Dr. Kumar is based at the New York University School of Medicine and is an affiliate professor at the University of Nairobi in Kenya.

**Author contribution.** All authors contributed to the writing of the original draft, review and editing.

**Financial support.** Preparation of this manuscript received no specific grant from any funding agency, commercial or not-for-profit sectors.

**Competing interests.** The authors declare none.

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
