## [Reviewer Report]

1. It was an absolute pleasure reviewing this manuscript. This is an important contribution to both advocacy and practice in the global mental health field.

2. Consider revising the title to: “An Expert Review of Major Implementation and Funding Challenges.” This may strengthen the clarity and positioning of the piece.

3. In the introduction, I recommend citing one of the authors’ earlier definitions of Global Mental Health, which is clear and concise:

“Global mental health is an evolving field of research and practice that aims to alleviate mental suffering through the prevention, care and treatment of mental and substance use disorders, and to promote and sustain the mental health of individuals and communities around the world.”

(Source: https://www.ncbi.nlm.nih.gov/pmc/articles/PMC7491634/)

4. Consider breaking up the following sentence for clarity, as it is quite long and dense:

“GMH includes universal access to effective prevention and treatment of mental health conditions and promotion of wellbeing, collectively referred to as Mental Health and Psychosocial Support (MHPSS)...”

Making this two sentences may improve readability and comprehension.

5. The opening section could be more concise. I recommend clearly distinguishing GMH and MHPSS early in the paper, especially since these terms are often used interchangeably but have distinct meanings. You mention the distinction in a footnote, but bringing it forward in the main text would provide helpful framing for the reader.

6. On Line 31, Page 3: The term “low-intensity psychosocial interventions” is not commonly used. WHO guidance more often refers to “psychological interventions.” Relevant resources include:

o https://iris.who.int/bitstream/handle/10665/376208/9789240087149-eng.pdf

o https://www.who.int/teams/mental-health-and-substance-use/treatment-care/innovations-in-psychological-interventions/

7. Consider including WHA Resolution WHA77.3 in the introduction. This recent and significant resolution affirms the prioritization of MHPSS in before, during and after emergencies and would strengthen the framing of the paper.

8. Line 38, Page 4: It may be helpful to define what is meant by a “standalone program” for readers who may not be familiar with the term.

9. Line 34, Page 5: The paragraph is content-rich but dense. Breaking it into smaller sections or clarifying its structure would improve readability.

10. Line 16, Page 6: Could you include an example of a digital mental health intervention that is evidence-based? This would ground the discussion in practical examples.

11. The recommendation section under “Support integration into physical health and other programs” could be strengthened by including the education system’s role in MHPSS, especially in humanitarian and development settings.

12. Line 44, Page 7: I suggest revising this sentence—

“MHPSS programs led by external, academic, or international non-governmental organizations (NGOs) instead tend to focus on reaching large numbers of relatively accessible people with interventions focused on prevention of single conditions (e.g. perinatal depression).”

—as it overlooks the fact that most psychological interventions are transdiagnostic and typically focus on common mental health conditions as a whole. The real gap tends to be in addressing severe mental health conditions like bipolar disorder and schizophrenia, for which less guidance and fewer resources exist.

13. Line 20, Page 8: Please clarify the phrasing—did you mean to write “very limited funds for MHPSS programming”?

14. The manuscript would benefit from a stronger recommendation on the treatment of severe mental health conditions, such as bipolar disorder and schizophrenia, which currently appears to be underemphasized.

15. Page 10: The examples of adaptation are very helpful. However, I suggest also noting why adaptation often doesn’t happen in practice—primarily due to lack of time, resources, and clear guidance. Many organizations default to generic manuals because there is no established process for contextual adaptation.

16. Page 11: Consider citing the locally developed CPT Functioning Tool as a best practice example:

https://www.elrha.org/news-blogs/cognitive-processing-therapy-functionality-tool

17. Page 14: The mhGAP Community Toolkit is a notable omission in this paragraph and should be referenced for completeness.

---

## [Reviewer Report]

Thankyou for the opportunity to review this paper. It addresses important issues, and consolidates the thinking of a group of authors around global MHPSS in a practical and helpful way. This paper has the potential to make a significant contribution to the field of global mental health.

A couple of aspects of the paper sit slightly uneasily with me, and I would like to highlight these as areas for the authors to reflect on.

Firstly, the paper is presented as though it is a broad consensus across the GMH field, yet it seems to be skewed towards US perspectives and universities in a way which does not reflect actual implementation of GMH in LMIC settings. Whilst this does not invalidate the arguments put forward in the paper, it would be helpful to make the point more clearly that this is a limited perspective. The authors do emphasise in the early part of the paper that they are focusing on only a specific type of GMH programme/ initiative, but as the paper progresses this point becomes lost and they appear to be referring to all GMH services. It would be helpful to return to the limitations of this review throughout the paper, especially when drawing conclusions.

Secondly, there are some developments in the MHPSS field which are not reflected in the paper. This gives the impression that the authors are not fully aware of the current state of the field, which I don’t think is the case - I suspect that the paper took a while to develop and the MHPSS field moved on somewhat in the meantime. In particular, the omission of the MHPSS Minimum Service Package stands out, especially in relation to the discussion of the need to integrate MHPSS into the work of other sectors. There is also the WHO resource on ‘Educating medical and nursing students in mental, neurological and substance use care - A WHO Guide to Pre-service Education’ which relates particularly to lines 33-40 on p12, and a whole range of resources on digital MHPSS for use in both emergency and development settings on the Red Cross Red Crescent Movement MHPSS Hub resource site (relevant to the second para on p6).

The recommendation that MH services are integrated into other sectors is the basis of the MHPSS in emergencies approach – in fact, one of the six core principles underpinning this approach is ‘integrated support systems’ (IASC (2007) Guidelines on MHPSS in Emergency Settings). So it would be worth noting that this recommendation is nothing new (the MHPSS MSP is all about this) but still needs to occur more systematically in practice.

A few more specific points:

P3, footnote. The term MHPSS was not first introduced by WHO – it was introduced through a consensus-based process involving many organisations, which resulted in the IASC (2007) Guidelines on MHPSS in Emergency Settings. The consensus base for the term is important, so it should be referenced using the IASC Guidelines.

P4, line 4: the abbreviation for psychosocial should be PS, not PSS (which is psychosocial support). This abbreviation probably isn’t needed since I don’t think it’s used again in the paper.

P9, line 46: a ref would be good for the statement that the literature suggests that the major approaches to treatment are effective across cultures.

P15, line 25: the word ‘needed’ at the beginning of the line should be deleted.

Finally, the end of the second para on p18 reads a bit strangely given recent developments within the US Govt in relation to humanitarian funding, but I imagine this paper was written before those events occurred.

---

## [Reviewer Report]

1. Page 11, line 339:

The reference stating, “The global mental health field has been described as one of north-to-south power imbalances and colonialist practices (Büyüm et al., 2020),” may be inaccurate. Upon reviewing the article, it appears that the authors are referring to global health more broadly, rather than specifically to global mental health.

I recommend either citing a source that directly addresses these dynamics within the global mental health field or revising the text to clarify that the reference pertains to global health broadly, with the implication that these dynamics may also apply to global mental health. Additionally, in the reference list, the name Büyüm should include the correct diacritical marks over the “ü’s.”

1. Pages 11 to 12:

One important point that seems to be missing is how uncommon it still is for organizations to adapt measurement tools to local contexts, even though the need for this is well recognized. This remains a major challenge across the field. There does not appear to be a widely accepted or viable alternative for either local or international organizations, especially given the pressure from donors to demonstrate impact through MEAL frameworks.

I suggest exploring this issue in more depth, particularly the tension between contextual relevance and donor-driven accountability. What is the way forward in a system where standardization is often prioritized over adaptation, even though local validity is essential?

3. General Comment:

Overall, I suggest going a bit deeper into the challenges you outline and the corresponding solutions. As global mental health practitioners, we are already well aware that these challenges exist. The pressing need now is to think more critically and concretely about the solutions, especially given the current funding crisis.

In this final version, I encourage you to explore more directly how to move beyond the status quo. For example, while it is widely accepted that mental health should be integrated into general health care, this often does not happen, even after training is completed. What are the deeper implementation challenges, and how might we begin to address them in more creative and practical ways?

---

## [Editor Report]

While the authors have adequately responded to one set of comments made by one reviewer, there are some minor clarifications sought by the other reviewer which should be addressed.